# TTF-1 Negativity Predicts Poor Outcomes in Advanced Non-Squamous NSCLC Also in the Immunotherapy Era: A Multicenter Cohort Study and Meta-Analysis

**DOI:** 10.3390/cancers17132188

**Published:** 2025-06-28

**Authors:** Leonardo Brunetti, Valentina Santo, Alessandro Galletti, Alain Gelibter, Antonio Lugini, Gian Paolo Spinelli, Daniele Santini, Alessio Cortellini, Alessia Vendittelli, Giuseppina Rita Di Fazio, Fabrizio Citarella, Giulia La Cava, Emanuele Claudio Mingo, Matteo Fiorenti, Leonardo Cristofani, Sabrina Mariotti, Serena Ricciardi, Francesco Pantano, Bruno Vincenzi, Giuseppe Tonini, Marco Russano

**Affiliations:** 1Department of Medical Oncology, Fondazione Policlinico Universitario Campus Bio-Medico, 200-00128 Rome, Italy; valentina.santo@unicampus.it (V.S.); agalletti2@scamilloforlanini.rm.it (A.G.); a.cortellini@policlinicocampus.it (A.C.); alessia.vendittelli@unicampus.it (A.V.); giuseppina.difazio@asst-fbf-sacco.it (G.R.D.F.); f.citarella@policlinicocampus.it (F.C.); giulia.lacava@unicampus.it (G.L.C.); e.mingo@unicampus.it (E.C.M.); matteo.fiorenti@unicampus.it (M.F.); f.pantano@policlinicocampus.it (F.P.); b.vincenzi@policlinicocampus.it (B.V.); g.tonini@policlinicocampus.it (G.T.); m.russano@policlinicocampus.it (M.R.); 2Department of Surgery and Cancer, Hammersmith Hospital Campus, Imperial College London, London SW7 2AZ, UK; 3Division of Medical Oncology, San Camillo Forlanini Hospital, 00152 Rome, Italy; sricciardi@scamilloforlanini.rm.it; 4Division of Medical Oncology B, Policlinico Umberto I, ‘La Sapienza’ University, 00161 Rome, Italy; alain.gelibter@uniroma1.it; 5Department of Oncology, San Giovanni-Addolorata Hospital, 00184 Rome, Italy; alugini@hsangiovanni.roma.it (A.L.); leonardo.cristofani@uniroma1.it (L.C.); 6UOC Territorial Oncology, UOS Experimental/IP—SL Latina—CdS, “Sapienza” University of Rome-Polo Pontino, 04011 Aprilia, Italy; gianpaolo.spinelli@uniroma1.it; 7UOC Oncologia A, Policlinico Umberto I, Sapienza University of Rome, 00161 Rome, Italy; daniele.santini@uniroma1.it; 8UOSD Oncologia Medica, Policlinico Tor Vergata, 00133 Rome, Italy; sabrina.mariotti@ptvonline.it

**Keywords:** NSCLC, immunotherapy, TTF-1, PD-L1

## Abstract

Lung cancer treatment has greatly improved thanks to immunotherapy, but not all patients respond equally. Currently, oncologists mainly use PD-L1 to decide who might benefit most from these treatments, but this is not always accurate. Our study examines another protein called Thyroid Transcription Factor-1 (TTF-1) to see if it could help predict outcomes in lung cancer patients treated with immunotherapy. We analyzed patient data from multiple hospitals and reviewed the existing scientific literature. We found that patients whose tumors did not produce TTF-1 had significantly worse outcomes, even if their PD-L1 levels were high. These results suggest that TTF-1 testing might be very useful to better select patients who need different or more aggressive treatment strategies. Including TTF-1 status in clinical practice could improve personalized treatment for lung cancer patients, but further studies are needed to confirm these findings.

## 1. Introduction

Despite several improvements in early diagnosis and anticancer treatments, lung cancer remains the leading cause of cancer-related mortality worldwide, with a considerable proportion of patients diagnosed at advanced stages and limited prognosis despite therapeutic advancements [1]. Adenocarcinoma represents the predominant histological subtype, and its management increasingly relies on molecular characterization to guide targeted treatments [2].

Immune checkpoint inhibitors (ICIs), specifically targeting the PD-1/PD-L1 axis, have fundamentally reshaped the therapeutic landscape of advanced non-small cell lung cancer (NSCLC). Landmark studies have established the superiority of immunotherapy alone or in combination with chemotherapy compared to chemotherapy alone, significantly improving survival outcomes [3,4,5,6]. Despite these advances, only a subset of patients achieves durable responses, highlighting the urgent need for more precise biomarkers beyond PD-L1 expression to predict immunotherapy efficacy and guide therapeutic decisions [7,8].

Thyroid Transcription Factor-1 (TTF-1, also known as NKX2-1) is a 38-kDa nuclear transcription factor primarily expressed in thyroid and lung epithelial tissues, where it regulates genes involved in cell differentiation, proliferation, and maintenance of pulmonary epithelial identity [9]. In NSCLC, particularly adenocarcinoma, TTF-1 expression has long been associated with favorable prognosis and enhanced sensitivity to pemetrexed-based chemotherapy [10,11]. This favorable clinical behavior may be biologically explained by the role of TTF-1 in maintaining a more differentiated tumor phenotype and modulating key signaling pathways related to tumor proliferation and immune microenvironment interactions [9,12].

Preliminary retrospective analyses have suggested a prognostic impact of TTF-1 expression even in the context of immunotherapy; however, results remain heterogeneous and have yet to be systematically validated [13,14]. Thus, a clear definition of TTF-1′s prognostic role in patients treated with first-line ICIs or chemo-immunotherapy is needed to refine patient stratification and treatment personalization strategies [15].

To address this clinical gap, our study employs a dual approach. Firstly, we conducted a multicenter retrospective analysis to explore the prognostic value of TTF-1 expression in patients with advanced non-squamous NSCLC receiving first-line immunotherapy or chemo-immunotherapy, and secondly, we performed a systematic literature review and meta-analysis to integrate our findings within the broader scientific context.

## 2. Materials and Methods

### 2.1. Study Design and Population

This is a retrospective, multicenter observational study. We enrolled 163 consecutive patients with histologically confirmed advanced non-squamous NSCLC treated between January 2020 and July 2024 across seven Italian oncology centers. Eligible patients included adults (≥18 years) diagnosed with stage IV non-squamous NSCLC, treated with first-line immunotherapy (IO) or chemo-immunotherapy (IO-CT). All 163 patients treated with first-line ICIs had been treated with pembrolizumab, while those treated with second-line ICIs received either nivolumab (n = 16), atezolizumab (n = 17), or pembrolizumab (n = 4). Chemotherapy regimens were restricted to pemetrexed plus platinum compounds (either cisplatin or carboplatin). Patients who relapsed after initial early-stage diagnosis and subsequently progressed to stage IV were also included. Only patients with measurable disease according to RECIST v1.1 criteria at baseline and evaluable TTF-1 expression were eligible.

To evaluate the predictive value of TTF-1 specifically in the immunotherapy era, an additional “historic” retrospective control cohort of 37 patients treated exclusively with first-line chemotherapy between April 2014 and September 2019 was also included, applying identical inclusion and exclusion criteria.

Exclusion criteria were the presence of actionable oncogenic driver mutations (EGFR, ALK, ROS-1, BRAF, NTRK, HER-2, RET, MET Ex14skip) and absence of measurable disease at baseline.

Radiological response assessments were routinely performed every 3 to 6 treatment cycles through computed tomography (CT), magnetic resonance imaging (MRI), or positron emission tomography/computed tomography (PET/CT), according to the investigator’s assessment.

### 2.2. Immunohistochemistry and Biomarker Assessment

TTF-1 expression was collected retrospectively from internal pathology reports, based on formalin-fixed, paraffin-embedded (FFPE) archival tumor specimens via immunohistochemistry (IHC). Nuclear staining for TTF-1 in ≥10% of tumor cells was considered positive, following widely accepted diagnostic criteria [16]. Local pathology laboratories conducted the immunostaining at the time of diagnosis using commercial antibodies (clone 8G7G3/1; Dako, Agilent Technologies, Santa Clara, CA, USA).

PD-L1 expression was evaluated on tumor cells using the 22C3 pharmDx assay (Dako, Agilent Technologies), categorized based on the tumor proportion score (TPS) into negative (<1%), intermediate (1–49%), and high expression (≥50%), according to standard clinical practice guidelines [7].

Clinical and pathological data, including age, gender, smoking history, performance status (ECOG PS), sites and number of metastases, treatment details, and radiological responses, were retrospectively collected from electronic medical records at participating institutions.

The study received approval from local institutional ethics committees and was conducted in accordance with the principles of the Declaration of Helsinki.

### 2.3. Statistical Analysis

Baseline patient characteristics were summarized using standard descriptive statistics. Categorical variables were compared using chi-square or Fisher’s exact tests, while continuous variables were compared using Mann–Whitney U tests. Survival analyses, including progression-free survival (PFS) and overall survival (OS), were conducted using Kaplan–Meier estimates and compared through log-rank tests. PFS was defined from the initiation of treatment until disease progression or death, whichever occurred first; OS from treatment initiation to death from any cause.

Univariate Cox proportional hazards models were initially employed to identify significant clinical parameters. Variables demonstrating statistical significance (*p* < 0.05) on univariate analysis were then included in multivariate Cox proportional hazards regression models. Interaction terms between TTF-1 and PD-L1 expression levels were tested to explore potential prognostic interdependence. Statistical analyses were performed using R software (version 4.4.3; R Core Team; R Foundation for Statistical Computing, Vienna, Austria, 2025).

### 2.4. Systematic Review and Meta-Analysis

A systematic review and meta-analysis were performed in accordance with PRISMA 2020 guidelines [17]. A systematic literature search was conducted according to the Preferred Reporting Items for Systematic Reviews and Meta-Analyses (PRISMA) guidelines. A comprehensive literature search was performed using PubMed and Europe PMC databases to identify studies evaluating the prognostic significance of Thyroid Transcription Factor-1 (TTF-1) in patients with advanced or metastatic non-small cell lung cancer (NSCLC) treated with immunotherapy. The search included studies published up to March 2025. In addition, we also employed an AI-based research assistant, Elicit (https://elicit.org, URL accessed on 27 March 2025), to identify potentially relevant studies that might not have been captured through conventional keyword-based queries. This hybrid approach was adopted to enhance the breadth and sensitivity of the evidence retrieval process. The workflow included three steps.

(1)A predefined search strategy on PubMed and Europe PMC using Boolean operators;(2)An AI-assisted search using Elicit across over 126 million academic papers from the Semantic Scholar corpus, in which the same PICO-based research question was posed;(3)A manual review and deduplication of all retrieved references from both sources.

Only studies that met the predefined eligibility criteria were included in the analysis. Discrepancies between traditional and AI-based outputs were resolved by consensus among reviewers.

Detailed search strategies, including specific search terms and Boolean operators, are provided in the Supplementary Methods (Table A1 and Figure A1).

Studies eligible for inclusion in the meta-analysis met the following criteria:

Clinical trials, retrospective or prospective studies evaluating advanced-stage (III/IV) NSCLC.Patients treated with immunotherapy (ICI) or immunotherapy combined with chemotherapy (IO-CT).Availability of hazard ratios (HRs) for PFS or OS stratified by TTF-1 expression.

Exclusion criteria were as follows:

Case reports, case series (<10 patients), reviews, editorials, or non-original studies.Preclinical studies or studies not clearly reporting outcomes stratified by TTF-1.Studies assessing only early-stage NSCLC or non-ICI-based therapies.

Two independent reviewers conducted study selection, and discrepancies were resolved by consensus or consultation with a third reviewer. Data extraction included study characteristics, patient demographics, treatment regimens, and outcome measures (HRs and 95% CIs). Meta-analysis was performed using random-effects models (DerSimonian–Laird method), and inter-study heterogeneity was assessed using the I^2^ statistic.

The studies included in the meta-analysis were as follows: Di Federico et al. [18], Galland et al. [14], Ibusuki et al. [19], Iso et al. [20], Ito et al. [21], Katayama et al. [22], Nakahama et al. [23], Nishioka et al. [24], and Moeller et al. [13].

## 3. Results

### 3.1. Patients’ Characteristics

A total of 200 patients with advanced non-squamous NSCLC were included in the study. Among them, 149 (74.5%) were classified as TTF-1-positive and 51 (25.5%) as TTF-1-negative. At the data cutoff (September 2024), 18% of patients were still progression-free, and 29.5% were alive.

Most patients were male (71%) and had a history of smoking (86%), with 80.5% presenting with de novo stage IV disease. The majority had ECOG performance status (PS) of 0–1 (99%), and adenocarcinoma was the predominant histological subtype (90.5%). Regarding PD-L1 expression, 27% had negative tumors, 44% had intermediate expression (1–49%), and 29% had high expression (≥50%).

Treatment distribution in the cohort was as follows: 106 patients (53%) received first-line chemo-immunotherapy (IO-CT), 57 (28.5%) received immune checkpoint inhibitors (ICIs) as monotherapy, and 37 (18.5%) were treated with chemotherapy alone. The overall objective response rate (ORR) and disease control rate (DCR) were 63% and 83.5%, respectively. Detailed baseline characteristics are presented in Table 1. 

When stratified by TTF-1 status (Table 2), the two groups were well balanced in terms of age, sex, smoking history, and performance status. As expected, TTF-1 expression was significantly associated with adenocarcinoma histology (*p* < 0.001). Notably, TTF-1-positive patients showed a higher frequency of baseline brain metastases (24.2% vs. 9.8%, *p* = 0.046). No significant differences were observed in the distribution of PD-L1 expression, treatment allocation, or metastatic burden. However, a higher disease control rate (86.6% vs. 70.6%, *p* = 0.015) was observed among TTF-1-positive patients, despite a non-significant difference in ORR (65.7% vs. 54.9%, *p* = 0.18).

### 3.2. Survival Analysis

At a median follow-up of 44.9 months, the median PFS and OS in the overall study population were 11.5 months (95% CI 9.7–15.3) and 21.8 months (95% CI 18.3–26.2), respectively.

When stratified by TTF-1 expression, patients with TTF-1-negative tumors exhibited significantly inferior outcomes. Median PFS was 6.7 months (95% CI 4.1–9.7) in the TTF-1-negative group compared to 16 months (95% CI 12.1–20.7) in the TTF-1-positive group (HR 2.22, 95% CI 1.59–3.13; *p* < 0.001). Similarly, median OS was markedly shorter in the TTF-1-negative group: 11.5 months (95% CI 9.8–17) versus 26.4 months (95% CI 22.4–33.3) in the TTF-1-positive group (HR 2.33, 95% CI 1.64–3.45; *p* < 0.001) (Figure 1a,b).

The prognostic relevance of TTF-1 expression remained consistent across different treatment subgroups. Among patients treated with immune checkpoint inhibitor monotherapy (IO), TTF-1 negativity was associated with a significantly worse PFS (median PFS 9.7 vs. 27.5 months; HR 2.33, 95% CI 1.09–5.00; *p* = 0.03). In the IO-CT group, TTF-1-negative patients had a median PFS of 6.3 months versus 12.8 months in the TTF-1-positive group (HR 1.92, 95% CI 1.22–3.03; *p* = 0.005). In the chemotherapy-only control cohort, the outcome gap appeared even more pronounced, with median PFS of 3.4 months for TTF-1-negative versus 13.2 months for TTF-1-positive patients (HR 3.33, 95% CI 1.52–7.14; *p* = 0.002) (Figure 2a–c).

### 3.3. Exploratory Analyses

#### 3.3.1. Interaction Between TTF-1 and PD-L1 Expression

To assess whether the prognostic impact of TTF-1 expression was modulated by PD-L1 expression levels, we tested a formal interaction between the two variables using a Cox proportional hazards model including an interaction term (TTF-1 × PD-L1, categorized as <50% vs. ≥50%). The analysis was restricted to patients treated with immunotherapy or chemo-immunotherapy as first-line treatment (n = 163), to ensure population homogeneity and avoid confounding due to differential treatment effects.

No statistically significant interaction was observed for PFS (HR for interaction term 1.17, 95% CI 0.67–2.03, *p* = 0.58) or OS (HR 1.02, 95% CI 0.58–1.79, *p* = 0.95), thus suggesting that TTF-1 and PD-L1 could retain independent prognostic value in this setting (Table A2).

Multivariable Cox regression analyses confirmed the independent prognostic value of TTF-1 negativity for both PFS and OS, even after adjusting for other baseline covariates. Specifically, in the PFS model, TTF-1 negativity (HR 2.17, 95% CI 1.49–3.13; *p* < 0.001), ECOG PS ≥1 (HR 1.49, 95% CI 1.11–2.01; *p* = 0.008), and PD-L1 expression ≥50% (HR 0.64, 95% CI 0.51–0.79; *p* < 0.001) were independently associated with outcome. Similar results were observed in the OS model, where TTF-1 negativity (HR 2.08, 95% CI 1.41–3.13; *p* < 0.001), ECOG PS ≥1 (HR 1.81, 95% CI 1.31–2.50; *p* < 0.001), and PD-L1 ≥50% (HR 0.61, 95% CI 0.49–0.77; *p* < 0.001) retained independent prognostic significance (Table 3).

#### 3.3.2. Combined Prognostic Value of TTF-1 and PD-L1 Expression

To further characterize the prognostic interplay between TTF-1 and PD-L1 expression, we constructed a four-level composite variable combining the two markers categorized as positive vs. negative for TTF-1 and high (≥50%) and low (<50%) depending on the PD-L1 expression: TTF-1−/PD-L1_low_, TTF-1−/PD-L1_high_, TTF-1+/ PD-L1_low_, and TTF-1+/ PD-L1_high_. In this model, the TTF-1−/ PD-L1_low_ group—hypothesized to represent the biologically most unfavorable phenotype—was used as the reference category.

In the Cox proportional hazards model for PFS, a significant reduction in risk of progression was observed only among TTF-1-positive patients. Specifically, TTF-1+/PD-L1_high_ patients exhibited the most favorable prognosis (HR 0.29, 95% CI 0.17–0.50, *p* < 0.001), followed by TTF-1+/PD-L1_low_ (HR 0.51, 95% CI 0.32–0.80, *p* = 0.004). Conversely, the TTF-1−/PD-L1_high_ group did not demonstrate a statistically significant survival advantage compared to the double-negative group (HR 0.63, 95% CI 0.30–1.35, *p* = 0.23).

Results were consistent for OS. The TTF-1+/PD-L1_high_ subgroup remained significantly associated with prolonged survival (HR 0.25, 95% CI 0.14–0.44, *p* < 0.001), as did the TTF-1+/PD-L1_low_ group (HR 0.53, 95% CI 0.33–0.84, *p* = 0.008). Again, the TTF-1−/PD-L1_high_ subgroup did not reach statistical significance (HR 0.62, 95% CI 0.28–1.37, *p* = 0.23).

These findings were further corroborated by stratified analyses conducted separately in TTF-1-positive and TTF-1-negative populations. Among TTF-1-positive patients, PD-L1 expression ≥50% was significantly associated with improved PFS (HR 0.56, 95% CI 0.35–0.87; *p* = 0.01) and OS (HR 0.46, 95% CI 0.27–0.76; *p* = 0.003). In contrast, no significant survival differences were observed according to PD-L1 status among TTF-1-negative patients, either in terms of PFS (HR 0.68, 95% CI 0.32–1.45; *p* = 0.31) or OS (HR 0.67, 95% CI 0.30–1.48; *p* = 0.32) (Table 4).

#### 3.3.3. Second-Line Outcomes and PFS2

At the time of data cutoff, a subset of 113 patients (56.5%) had received a second-line systemic treatment following disease progression. Among these, 30 were TTF-1-negative and 83 were TTF-1-positive. Second-line treatments included either chemotherapy or immune checkpoint inhibitors (ICIs). In total, 36.7% of TTF-1-negative patients and 31% of TTF-1-positive patients received ICIs, while the remainder received chemotherapy.

Baseline characteristics between TTF-1-positive and -negative patients within the second-line population were balanced (Table A3). Kaplan–Meier analysis revealed no statistically significant difference in second-line progression-free survival (PFS2) between the two groups. Median PFS2 was 4.35 months (95% CI 3.00–9.43) in TTF-1-negative patients and 4.17 months (95% CI 3.47–6.20) in TTF-1-positive patients. The corresponding HR was 1.23 (95% CI 0.81–1.89; *p* = 0.34) (Figure A2).

### 3.4. Meta-Analysis of the Prognostic Role of TTF-1 Expression in Advanced NSCLC

To contextualize our findings, we conducted a meta-analysis of published studies evaluating the prognostic role of TTF-1 expression in patients with advanced non-squamous NSCLC treated with immunotherapy-based regimens. A total of 14 cohorts from 9 studies were included for the PFS endpoint. For OS, the final analysis encompassed 13 cohorts after excluding Katayama et al. [22], which did not report OS data. All studies were retrospective, with sample sizes ranging from 58 to 755 patients. The majority investigated first-line treatment, except for a few that included second or subsequent lines. TTF-1 positivity was defined variously—most frequently as any nuclear staining (e.g., “focal positivity”) or a >10% threshold—while PD-L1 was generally assessed by 22C3 pharmDx (in one case combined with “QR1”) and reported in categories (<1%, 1–49%, ≥50%). Median age across cohorts ranged roughly from 63 to 73 years, and all studies reporting stage of diagnosis enrolled patients with stage III–IV disease. Additional details on study design, PD-L1 clone, line of therapy, and TTF-1 criteria are summarized in Table A4.

Baseline characteristics for each included cohort, along with the key outcomes (HRs for PFS and OS), are provided in Table 5.

Among the 14 cohorts, median PFS across the studies ranged from 1.6 to 9.9 months in the TTF-1-negative groups and from 2.6 to 12.2 months in the TTF-1-positive groups. Median OS ranged from 5.7 to 42.5 months, depending on treatment type and TTF-1 status. The largest study (Di Federico et al. [18]) contributed two large cohorts (755 patients IO; 294 patients IO-CT).

In the overall analysis, TTF-1 negativity was significantly associated with worse PFS, with a pooled HR of 1.75 (95% CI 1.50–2.04, *p* < 0.0001; I^2^ = 29.6%) (Figure 3a). Subgroup analysis revealed that the effect was robust across treatment modalities, with pooled HRs of 1.96 (95% CI 1.54–2.49) for IO-treated patients, 1.63 (95% CI 1.25–2.13) for IO-CT-treated patients, and 1.32 (95% CI 0.86–2.02) for mixed regimens. Between-subgroup heterogeneity was not statistically significant (*p* = 0.18) (Figure 3a).

Similarly, for OS, TTF-1 negativity was associated with significantly worse outcomes, with a pooled HR of 1.76 (95% CI 1.45–2.14, *p* < 0.0001; I^2^ = 65.1%) (Figure 3b). The detrimental impact was consistent within IO-treated (HR 1.93, 95% CI 1.28–2.92) and IO-CT-treated patients (HR 1.73, 95% CI 1.42–2.12), while the single mixed study showed a similar but non-significant trend (HR 1.25, 95% CI 0.83–1.87). Again, there was no significant subgroup difference (*p* = 0.24) (Figure 3b).

Importantly, our study contributed data to both treatment subgroups. Among patients treated with IO (n = 57), the reciprocal HRs were 2.33 (95% CI 1.00–5.44) for PFS and 2.38 (95% CI 1.04–5.43) for OS, while for the IO-CT cohort (n = 106), HRs were 1.92 (95% CI 1.22–3.03) for PFS and 1.92 (95% CI 1.25–2.95) for OS. These results were directionally concordant with the pooled estimates and further support the unfavorable prognostic impact of TTF-1 loss.

## 4. Discussion

Thyroid transcription factor-1 (TTF-1) has long been recognized as a marker of lung adenocarcinoma, and retrospective studies have hinted that TTF-1-negative tumors might achieve inferior responses to chemotherapy and, possibly, immunotherapy [11,19]. However, the magnitude and consistency of TTF-1′s prognostic impact in the modern immunotherapy era remain incompletely defined, particularly across heterogeneous patient populations and treatment lines.

### 4.1. Main Findings from Our Retrospective Cohort

In our multicenter retrospective cohort of advanced non-squamous NSCLC (n = 200), TTF-1 negativity emerged as a strong and independent predictor of unfavorable prognosis. The median PFS for TTF-1-negative tumors ranged from 3.4 months under chemotherapy alone to 9.7 months under immunotherapy alone (IO), whereas TTF-1-positive patients achieved 11.0 and 27.5 months, respectively, in the same treatment cohorts. A similar gap was observed in OS, with TTF-1-negative tumors reaching a median of 11.5 months compared to 26.4 months for TTF-1-positive tumors. These differences persisted in multivariable analyses, underscoring TTF-1′s robust prognostic role even after adjusting for PD-L1 expression and ECOG performance status.

Notably, TTF-1-negative patients still had worse outcomes despite a slightly higher proportion of PD-L1 ≥50% (31.4% vs. 25.5% in TTF-1-positive), suggesting that TTF-1 negativity can override or diminish the usual positive impact associated with high PD-L1. Interestingly, brain metastases were more common among TTF-1-positive patients (24.2% vs. 9.8%), despite their overall more favorable prognosis, although this did not significantly impact survival in multivariable models in our cohort, this effect could have been mitigated by the fact that TTF-1-negative brain metastases are already demonstrated to result in worse prognosis when compared to TTF-1-positive ones [25,26]. By contrast, the prognostic gap was relatively attenuated in the chemo-immunotherapy (IO-CT) combination group (6.3 vs. 12.8 months of PFS for TTF-1-negative vs. -positive), indicating that dual-modality treatment may partially mitigate—but not eliminate—the adverse effect of TTF-1 loss.

Beyond the first line setting, no meaningful differences in second-line PFS2 were observed. The TTF-1-negative group had a median PFS2 of 4.35 months, compared to 4.17 months in TTF-1-positive patients. Several factors might contribute to this result, including a more heterogeneous mix of second-line regimens and potential survivor bias (only patients fit enough to receive further therapy). Nonetheless, our data overall show that TTF-1 negativity confers a consistent negative prognostic impact in the first-line scenario, particularly when immunotherapy or chemotherapy is used alone, and that PD-L1 status does not substantially mitigate this effect.

### 4.2. TTF-1 vs. PD-L1 as Prognostic Biomarkers

Our results also highlight that TTF-1 negativity is a stronger determinant of poor prognosis than low PD-L1 expression. Although PD-L1 ≥50% remains a well-established biomarker for immunotherapy efficacy [3,4], its prognostic power can be less clear, especially in real-world populations with diverse lines of therapy [8]. Indeed, TTF-1-negative tumors in our study fared poorly even in the presence of high PD-L1. These findings align with smaller retrospective reports [14,19,23] describing TTF-1 negativity as indicative of a more aggressive tumor biology and diminished immunotherapy benefit. A possible explanation is that TTF-1-negative adenocarcinomas are often less differentiated and may harbor molecular or immunologic features that evade immune surveillance, regardless of PD-L1 level. Several reports suggest that TTF-1 loss may mark a biologically distinct subtype of NSCLC with lower differentiation, higher histologic grade, and reduced expression of genes involved in antigen presentation and T cell recruitment [27]. This could foster an immune-excluded tumor microenvironment and explain reduced benefit from ICIs [28,29]. These findings are supported by genomic and transcriptomic analyses highlighting lower TMB and reduced immune gene signatures in TTF-1-negative tumors [8,9,30].

### 4.3. Insights from the Meta-Analysis

To contextualize our findings within the broader literature, we performed a systematic review and meta-analysis of studies investigating TTF-1 expression as a prognostic biomarker in advanced NSCLC patients receiving immunotherapy-based regimens. This meta-analysis is, to our knowledge, the first quantitative synthesis focusing specifically on TTF-1 in the modern immunotherapy era. Definitions of TTF-1 positivity varied across included studies, ranging from any detectable nuclear staining to a ≥10% threshold, while many studies did not report the staining method used for the classification. This variability could influence the pooled estimates and should be considered when interpreting the findings. Despite heterogeneity in TTF-1 assessment (e.g., >10% threshold vs. any focal positivity) [31] and variations in treatment lines, our pooled results consistently showed that TTF-1 negativity is associated with significantly worse outcomes for both PFS and OS. In line with our own data, these detrimental effects proved robust across different immunotherapy approaches—IO alone or IO-CT—and persisted in subgroup analyses.

Notably, all included studies in the meta-analysis were retrospective, some with relatively small sample sizes or single-center designs [13,14,20]. Nevertheless, even the largest dataset [14] upheld the negative impact of TTF-1 loss. Overall, these findings reinforce TTF-1 negativity’s relevance across varying populations and lines of therapy, although definitive prospective validation remains lacking.

### 4.4. Limitations

Both our retrospective cohort and the meta-analysis share important limitations. First, the observational design predisposes to selection biases and residual confounding, despite our efforts to adjust for baseline factors such as ECOG PS and PD-L1. Second, our subgroup sizes—particularly TTF-1-negative patients receiving IO alone—are modest, reducing statistical power. Third, TTF-1 staining protocols varied among centers; while we aimed for a ≥10% nuclear positivity threshold, minor differences in interpretation are possible, also given the lack of reporting in several studies included in the meta-analysis. Fourth, while most studies employed the 22C3 clone for PD-L1 evaluation, some (e.g., Di Federico et al. [18], Ibusuki et al. [19]) did not report the antibody used. This lack of standardization may contribute to variability in PD-L1 stratification. Fifth, the meta-analysis included only retrospective studies, and certain HRs (Galland et al. [14]) were extracted using WebPlotDigitizer or lacked multivariable adjustment. Finally, we did not conduct line-specific sensitivity analyses in the meta-analysis, and some cohorts combined first-line with later-line immunotherapy, potentially inflating heterogeneity.

### 4.5. Clinical Implications

Collectively, these findings suggest that TTF-1 status should be considered a key prognostic biomarker in advanced NSCLC, potentially rivalling PD-L1 in importance for outcome prediction. Indeed, TTF-1 negativity highlights a high-risk subset who may warrant intensified therapy or closer monitoring. Furthermore, TTF-1 could serve as a valuable stratification factor in prospective trials, ensuring balanced arms for a variable strongly linked to survival outcomes. As immunotherapy research evolves to incorporate novel agents—such as dual immune checkpoint blockade or targeted immunomodulators—accounting for TTF-1 in trial design may prevent imbalances in baseline risk.

## 5. Conclusions

Our findings confirm that TTF-1 negativity stands out as a significant and independent adverse prognostic factor in advanced non-squamous NSCLC, persisting across multiple therapeutic modalities. The meta-analysis reinforces its negative impact on PFS and OS, supporting the need to consider this factor within clinical trials and future prospective studies, particularly given that, according to the literature, approximately 15–20% of non-squamous NSCLCs are TTF-1-negative at diagnosis. By systematically evaluating TTF-1 expression alongside PD-L1 and other established biomarkers, clinicians and researchers may refine patient selection and ultimately improve outcomes, particularly for those TTF-1-negative tumors that appear resistant to standard immunotherapy regimens.

## Figures and Tables

**Figure 1 cancers-17-02188-f001:**
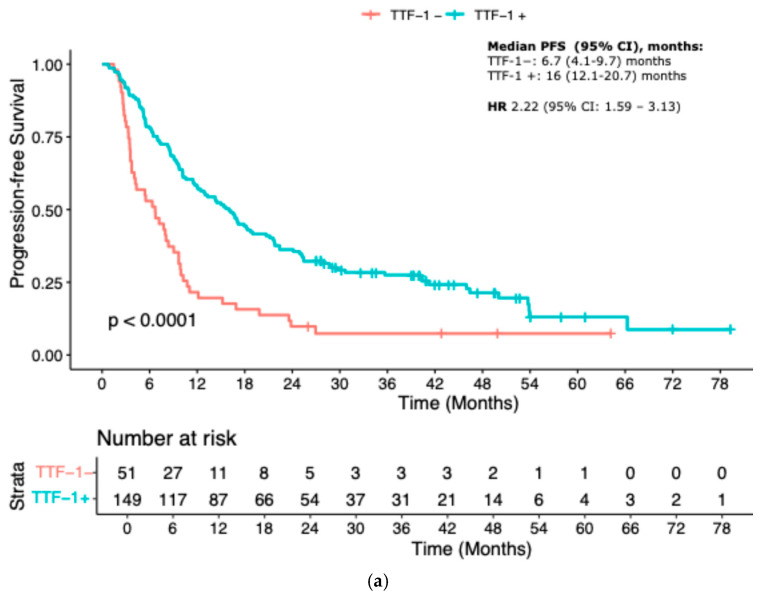
Kaplan–Meier curves according to TTF-1 status: (**a**) Progression-free Survival; (**b**) Overall survival.

**Figure 2 cancers-17-02188-f002:**
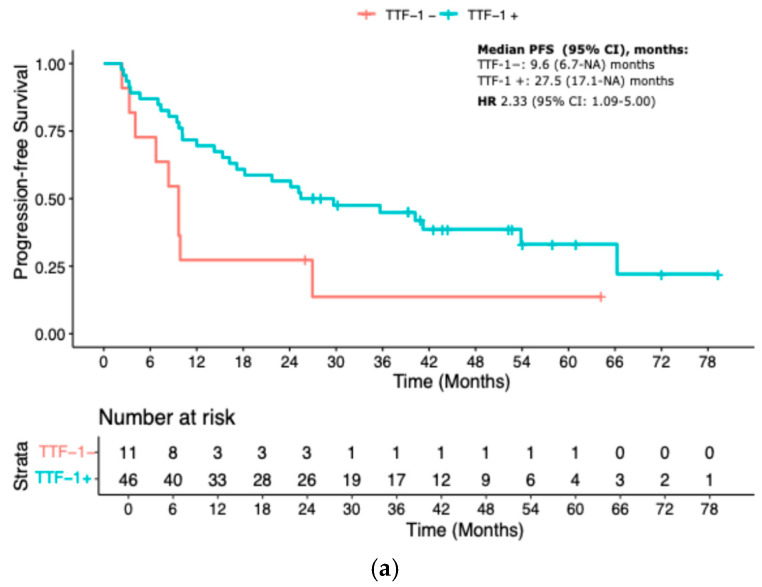
KM curves for PFS according to treatment. (**a**) IO-mono; (**b**) Chemo-IO; (**c**) Chemotherapy. IO-mono = immune checkpoint inhibitor monotherapy; Chemo-IO = chemo-immunotherapy.

**Figure 3 cancers-17-02188-f003:**
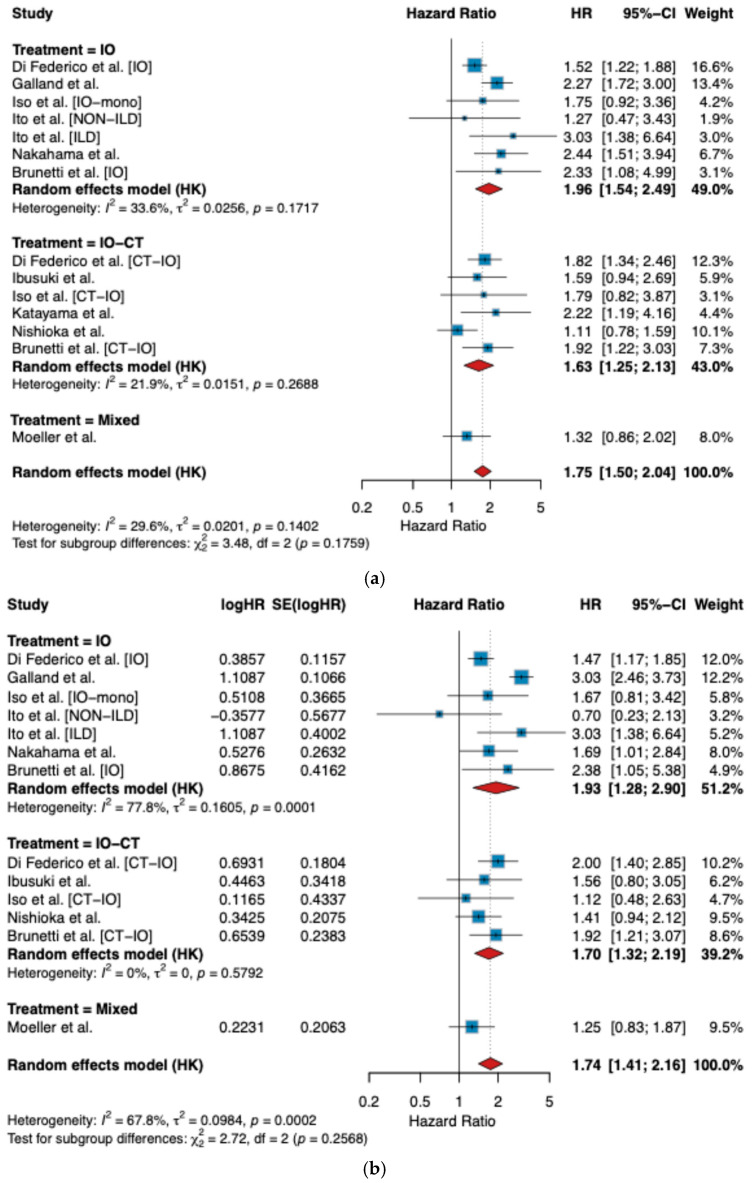
Forest plot for TTF-1 impact on survival: (**a**) PFS; (**b**) OS [13,14,18,20,21,22,23,24].

**Table 1 cancers-17-02188-t001:** Baseline demographic data and main characteristics of the overall population.

Characteristics		Total N 200 (100%); Median (Q1–Q3)
Median Age		67 (60–73)
Sex	Women	58 (29%)
	Men	142 (71%)
Smoking Habit	Never Smoker	28 (14%)
	Former Smoker	97 (48.5%)
	Heavy Smoker	75 (37.5%)
Stage at Diagnosis (AJCC 8th Edition)	IA/IB	4 (2%)
	IIA	6 (3%)
	IIB	2 (1%)
	IIIA	17 (8.5%)
	IIIB	7 (3.5%)
	IIIC	3 (1.5%)
	IV	161 (80.5%)
PS ECOG	0	107 (53.5%)
	1	91 (45.5%)
	2	2 (1%)
PD-L1 Expression	0	54 (27%)
	1–49%	88 (44%)
	>50%	58 (29%)
TTF-1 Expression	Negative	51 (25.5%)
	Positive	149 (74.5%)
Histology	Adenocarcinoma	181 (90.5%)
	NOS	19 (9.5%)
First Line Treatment	Immunotherapy	57 (28.5%)
	Immuno-chemotherapy	106 (53%)
	Chemotherapy	37 (18.5%)
	Pembrolizumab	167 (83.5%)
PD(L)-1 antibody	Atezolizumab	17 (8.5%)
	Nivolumab	16 (8%)

**Table 2 cancers-17-02188-t002:** Patients’ characteristics by TTF-1 expression.

Patients’ Characteristics by TTF-1 Status-n (%); Median (Q1–Q3)				
		TTF-1− (n = 51)	TTF-1+ (n = 149)	*p*-Value
Sex	Men	15 (29.4%)	43 (28.9%)	*p* > 0.9
	Women	36 (71.6%)	106 (71.1%)	
Age, years	≥70	24 (47.1%)	61 (40.9%)	*p* = 0.5
	<70	27 (52.9%)	88 (59.1%)	
Smoking habit	Never Smoker	6 (11.8%)	22 (14.8%)	
	Former Smoker	27 (52.9%)	70 (47%)	*p* = 0.7
	Heavy Smoker	18 (35.3%)	57 (38.2%)	
Histology	Adenocarcinoma	40 (78.4%)	141 (94.6%)	** *p* ** ** < 0.001**
	NOS	11 (21.6%)	8 (5.4%)	
PS ECOG	0	24 (47%)	83 (55.7%)	
	1	26 (51%)	65 (43.6%)	*p* = 0.4
	2	1 (2%)	1 (0.7%)	
Brain Metastases	Yes	5 (9.8%)	36 (24.2%)	** *p* ** ** = 0.046**
	No	46 (90.2%)	113 (75.8%)	
Bone Metastases	Yes	22 (43.1%)	50 (33.6%)	*p* = 0.3
	No	29 (56.9%)	99 (66.4%)	
Liver Metastases	Yes	7 (13.7%)	17 (11.4%)	*p* = 0.8
	No	44 (86.3%)	132 (88.6%)	
Metastatic Burden	≥3	21 (41.2%)	61 (40.9%)	*p* > 0.9
	<3	30 (58.8%)	88 (59.1%)	
PDL-1 expression	Negative	12 (23.5%)	46 (31%)	
	1–49%	23 (45.1%)	65 (43.5%)	*p* = 0.5
	≥50%	16 (31.4%)	38 (25.5%)	
First line treatment	IO	11 (21.5%)	46 (31%)	
	IO-CT	29 (57%)	77 (51.7%)	*p* = 0.4
	CT	11 (21.5%)	26 (17.3%)	
Best Response	CR	-	6 (4%)	** *p* ** ** = 0.038**
	PR	28 (54.9%)	92 (61.7%)	
	SD	8 (15.7%)	31 (20.8%)	
	PD	15 (29.3%)	20 (13.5%)	
ORR		54.9%	65.7%	*p* = 0.18
DCR		70.6%	86.6%	** *p* ** ** = 0.015**

**Table 3 cancers-17-02188-t003:** Uni and multivariate Cox regressions for PFS and OS respectively.

**PFS**	**Univariate**		**Multivariate**	
**Covariates**	**Hazard Ratio (95% CI)**	** *p* ** **-Value**	**Hazard Ratio (95% CI)**	** *p* ** **-Value**
TTF-1 (Negative vs. Positive)	**2.22 (1.59–3.13)**	**<0.001**	**2.17 (1.49–3.13)**	**<0.001**
Sex (Female vs. Male)	0.89 (0.64–1.26)	0.52	0.89 (0.63–1.27)	0.53
Histology (Adc vs. non-Adc)	0.95 (0.55–1.61)	0.83	0.85 (0.47–1.52)	0.57
Brain Metastases (Yes vs. no)	0.98 (0.67–1.43)	0.9	1.07 (0.72–1.76)	0.72
PS ECOG (≥1 vs. 0)	**1.44 (1.07–1.92)**	**0.02**	**1.49 (1.11–2.01)**	**0.008**
Smoking History (Yes vs. no)	0.99 (0.79–1.24)	0.9	1.09 (0.85–1.39)	0.49
PD-L1 (≥50% vs. <50%)	**0.67 (0.55–0.83)**	**<0.001**	**0.64 (0.51–0.79)**	**<0.001**
Age (≥70 vs. <70 years)	1.15 (0.85–1.57)	0.36	1.11 (0.8–1.54)	0.52
OS	**Univariate**		**Multivariate**	
**Covariates**	**Hazard Ratio (95% CI)**	** *p* ** **-Value**	**Hazard Ratio (95% CI)**	** *p* ** **-Value**
TTF-1 (Negative vs. Positive)	**2.33 (1.64–3.45)**	**<0.001**	**2.08 (1.41–3.13)**	**<0.001**
Sex (Female vs. Male)	0.88 (0.61–1.28)	0.51	0.9 (0.62–1.31)	0.59
Histology (Adc vs. non-Adc)	0.79 (0.46–1.38)	0.41	0.65 (0.35–1.2)	0.17
Brain Metastases (Yes vs. no)	1.05 (0.69–1.58)	0.82	1.19 (0.77–1.81)	0.43
PS ECOG (≥1 vs. 0)	**1.74 (1.27–2.4)**	**<0.001**	**1.81 (1.31–2.5)**	**<0.001**
Smoking History (Yes vs. no)	0.94 (0.741–1.2)	0.63	1.04 (0.81–1.33)	0.77
PD-L1 (≥50% vs. <50%)	**0.68 (0.54–0.85)**	**<0.001**	**0.61 (0.49–0.77)**	**<0.001**
Age (≥70 vs. <70 years)	1.34 (0.96–1.87)	0.08	1.27 (0.89–1.81)	0.19

**Table 4 cancers-17-02188-t004:** Median PFS and OS for PD-L1 high vs. low in TTF-1- and TTF-1+ patients.

		PFS		OS	
	PD-L1	Months (95% CI)	HR (95% CI) *p*-Value	Months (95% CI)	HR (95% CI) *p*-Value
TTF-1 -	High (≥50%)	9.67 (6.73-NA)	0.68 (0.32–1.45) *p* = 0.313	13.5 (6.73-NA)	0.67 (0.3–1.48) *p* = 0.322
	Low (<50%)	6.33 (4.2–10.2)		11 (6.2–19.3)	
TTF-1 +	High (≥50%)	25.3 (16.2-NA)	**0.56 (0.35–0.87)** ***p* = 0.011**	NA (26.2-NA)	**0.46 (0.27–0.76)** ***p* = 0.003**
	Low (<50%)	13 (9.6–21.2)		22.4 (17.8–28)	

**Table 5 cancers-17-02188-t005:** Overview of studies included in the meta-analysis. * Reconstructed with WebPlotDigitizer.

Author (year)	Intervention Type	Cases	Median Follow-Up	Median OS	Median PFS	HR OS	HR PFS
Di Federico et al., 2023 [18]	IO	755	NR	13.5 vs. 6.3 months	3.5 vs. 2.1 months	0.68 (0.54–0.85)	0.66 (0.53–0.82)
	IO-CT	294	-	20 vs. 7.6 months	7 vs. 3.8 months	0.5 (0.35–0.71)	0.55 (0.41–0.75)
Galland et al., 2021 [14]	IO	231	NR	NR	NR	0.33 (0.27–0.41) *	0.44 (0.35–0.61) *
Ibusuki et al., 2022 [19]	IO-CT	122	14.6 months	10.8 vs. 5.7 months	12.2 vs. 6 months	0.64 (0.33–1.26)	0.63 (0.37–1.06)
Iso et al., 2023 [20]	IO	70	17.9 months	19.5 vs. 15.6 months	4.5 vs. 3.8 months	0.6 (0.29–1.22)	0.57 (0.3–1.1)
	IO-CT	59	-	18.9 vs. 32.3 months	9.6 vs. 9.9 months	0.89 (0.38–2.08)	0.56 (0.26–1.22)
Ito et al., 2025 [21]	IO (non-ILD)	28	NR	37.1 vs. 33.7 months	2.6 vs. 1.8 months	1.43 (0.47–4.35)	0.79 (0.29–2.13)
	IO (ILD)	34	-	42.5 vs. 14.5 months	12 vs. 2 months	0.33 (0.15–0.72)	0.33 (0.15–0.72)
Nakahama et al., 2022 [23]	IO	108	NR	18.2 vs. 8 months	5.4 vs. 1.6 months	0.59 (0.36–1.01)	0.41 (0.26–0.68)
Katayama et al., 2023 [22]	IO-CT	58	10.9 months	-	10.9 vs. 5 months	-	0.45 (0.24–0.84)
Nishioka et al., 2025 [24]	IO-CT	190	21.3	25 vs. 21.2 months	7.6 vs. 6 months	0.71 (0.47–1.06)	0.9 (0.63–1.29)
Moeller et al., 2022 [13]	IO + IO-CT	154	NR	8.4 vs. 5.8 months	6.5 vs. 4.6 months	0.8 (0.53–1.19)	0.76 (0.51–1.2)

## Data Availability

The datasets presented in this article are not readily available due to privacy reasons. Requests to access the datasets should be directed to the authors.

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
