# Peer review of "TTF-1 Negativity Predicts Poor Outcomes in Advanced Non-Squamous NSCLC Also in the Immunotherapy Era: A Multicenter Cohort Study and Meta-Analysis"

_cancers, 2025, doi:10.3390/cancers17132188_

Round 1
Reviewer 1 Report
Comments and Suggestions for Authors
This is a nice manuscript discussing lung cancer immunohistochemical profiling of TTF-1 in the prognosis of the disease.
The authors have followed the guidelines for meta-analyses and included large studies on NSCLC patients at Stages 3,4.
A few comments to be addressed:
- The authors claim to have included advanced lung cancer cases. But in Table 1 they clearly present cases of Stage 1,2 that were included in the analysis.
- The "traditional cohort" that was included is earlier than 2017 that IASCLC was published. Please check for integrity of definitions.
- AI tools from outsourced private (non-Academic) company were used. Please explain according to the Journal guidelines.
Author Response
Thank you for your comments, all comments have been addressed and modification to the text have been made accordingly.
Comment 1: the authors claim to have included advanced lung cancer cases. But in Table 1 they clearly present cases of Stage 1,2 that were included in the analysis.
Reply 1: All of the patients were included in the study when diagnosed at an advanced stage. The table 1 data refers to the initial diagnosis, e.g. if the patients were diagnosed as upfront metastatic NSCLC or if they suffered from a later progression after an initial diagnosis at an early stage.
Comment 2: The "traditional cohort" that was included is earlier than 2017 that IASCLC was published. Please check for integrity of definitions.
Reply 2: It's true that these patients come from an earlier period than the AJCC 2017 edition, but all of them were included in the study starting from the advanced stage diagnosis, which is consistent with previous definitions.
Comment 3: AI tools from outsourced private (non-Academic) company were used. Please explain according to the Journal guidelines
Reply 3: As stated in the "acknowledgments" paragraph, the AI-based Elicit tool was used to aid in the generation of the library for the metanalysis part, having generated a comprehensive report from the Semantic Scholar library, contributing with 499 text that were subsequentely screened and assessed together with all of the others text.
Reviewer 2 Report
Comments and Suggestions for Authors
Brunetti and colleagues, combining a multicenter retrospective cohort with a thoughtfully designed meta-analysis demonstrate association between TTF-1 negativity and worse outcomes, also including patients with high PD-L1 expression, highlighting the need to incorporate TTF-1 into clinical risk stratification.The manuscript is generally clear and well written with robust analysis, and high translational value. The inclusion of both real-world data and a quantitative synthesis of the literature strengthens the conclusions.
I have a few minor considerations which should be addressed to further improve the manuscript:
- The imbalance in brain metastases (higher in TTF-1 positive patients) may influence survival outcomes which could further impaired by the relatively small size(25.5%) of TTF-1 negative subgroup as a whole, this could affect the power of some analyses.Consider adjusting or stratifying for this in additional multivariate analyses or including a note on its potential impact in the discussion. I would also ad that daily clinical practice suggest brain metastasis are often TTF1 negative, please add a brief comment.
A brief sensitivity analysis or stratified comment in the discussion regarding variability in TTF1 positivity definitions (>10% nuclear staining vs. any focal) would be helpful to contextualize the robustness of the pooled results and elucidate staining method variability.
- Heterogeneity across studies in TTF-1 staining thresholds and PD-L1 clones (not reported in a few studies such as Di Federico and Ibusuki-..) could influence pooled results, a sensitivity analyses stratified by staining method or PD-L1 assay type could and should increase transparency.
- The discussion could explore potential biological mechanisms behind TTF-1's effect on immune response more deeply especially elucidating the link between it's expression and mutational burden and cellular grading with could elicit immune evasion pathways.
Please add a bit more detail to figure legends, particularly Kaplan–Meier curves, so they can be interpreted standalone (e.g., define IO and IO-CT in the legend itself).
Meta-Analysis Limitations: Mention explicitly that some HRs were reconstructed via WebPlotDigitizer, and consider stating which studies lacked multivariable adjustment.
Typographical Errors: Minor typos (e.g., “ackowledge” instead of “acknowledge” in the Acknowledgments) should be corrected.
Author Response
Comment 1: The imbalance in brain metastases (higher in TTF-1 positive patients) may influence survival outcomes which could further impaired by the relatively small size(25.5%) of TTF-1 negative subgroup as a whole, this could affect the power of some analyses. Consider adjusting or stratifying for this in additional multivariate analyses or including a note on its potential impact in the discussion
Reply 1: We thank the reviewer for this thoughtful observation. Indeed, brain metastases were more frequent among TTF-1 positive patients in our cohort (24.2% vs 9.8%), which is seemingly counterintuitive given their overall more favorable outcomes. We have now added a comment in the Discussion highlighting this finding as potentially reflective of distinct metastatic patterns, in contrast with clinical experience suggesting that brain lesions are frequently TTF-1 negative when tested. As per the clinical practice suggestion, while we agree with the sensation that brain metastases can often result as TTF-1 negative even with TTF-1 positive primaries, there is no confirmation in literature of such intuition to best of our knowledge, while it could be surely room for further research.
Comment 2: A brief sensitivity analysis or stratified comment in the discussion regarding variability in TTF1 positivity definitions (>10% nuclear staining vs. any focal) would be helpful to contextualize the robustness of the pooled results and elucidate staining method variability.
Reply 2: We agree with the reviewer that the definition of TTF-1 positivity varies across studies. While our study adopted a ≥10% nuclear staining threshold, others considered any focal nuclear staining as positive. We have now added a paragraph in the Discussion to acknowledge this heterogeneity and its potential implications for interpretation of the meta-analytic results, while we already addressed this in the limitations paragraph. Unfortunately, we think a sensitivity analysis would be too limited in value, given the lack of reporting for the staining method from several studies, which could impact the results of such analysis.
Comment 3: Heterogeneity across studies in TTF-1 staining thresholds and PD-L1 clones (not reported in a few studies such as Di Federico and Ibusuki-..) could influence pooled results, a sensitivity analyses stratified by staining method or PD-L1 assay type could and should increase transparency.
Reply 3: We thank the reviewer for this important point. We have now included a note in the Limitations section stating that a few studies (e.g., Di Federico et al., Ibusuki et al.) did not report the PD-L1 clone used. Although a formal subgroup analysis was not feasible, this heterogeneity may have contributed to residual variability in the pooled results.
Comment 4: The discussion could explore potential biological mechanisms behind TTF-1's effect on immune response more deeply especially elucidating the link between it's expression and mutational burden and cellular grading with could elicit immune evasion pathways.
Reply 4: We appreciate this insightful suggestion. We have now expanded the Discussion to better explore the biological rationale for the poor outcomes observed in TTF-1 negative tumors. In particular, we propose that TTF-1 loss may reflect a dedifferentiated tumor phenotype characterized by reduced lineage identity, higher histological grade, impaired antigen presentation, and immune-cold features. These traits could contribute to diminished immunotherapy efficacy. We have also added literature citations that support these hypotheses.
Comment 5-7: Please add a bit more detail to figure legends, particularly Kaplan–Meier curves, so they can be interpreted standalone (e.g., define IO and IO-CT in the legend itself). Meta-Analysis Limitations: Mention explicitly that some HRs were reconstructed via WebPlotDigitizer, and consider stating which studies lacked multivariable adjustment. Typographical Errors: Minor typos (e.g., “ackowledge” instead of “acknowledge” in the Acknowledgments) should be corrected.
Reply 5-7: figure legends have been updated to include definitions of IO and IO-CT. We now explicitly mention the use of WebPlotDigitizer for HR extraction from Kaplan–Meier curves in the Meta-analysis and Limitations sections. All typographical errors noted by the reviewer have been corrected.